# Retinopathy of Prematurity and MicroRNAs

**DOI:** 10.3390/biomedicines13020400

**Published:** 2025-02-07

**Authors:** Giuseppe Maria Albanese, Giacomo Visioli, Ludovico Alisi, Marta Armentano, Francesca Giovannetti, Luca Lucchino, Marco Marenco, Paola Pontecorvi, Magda Gharbiya

**Affiliations:** 1Department of Sense Organs, Sapienza—University of Rome, Viale del Policlinico 155, 00161 Rome, Italy; giuseppemaria.albanese@uniroma1.it (G.M.A.); magda.gharbiya@uniroma1.it (M.G.); 2Policlinico Umberto I University Hospital, Viale del Policlinico 155, 00161 Rome, Italy; 3Department of Experimental Medicine, Sapienza—University of Rome, Viale del Policlinico 155, 00161 Rome, Italy; paola.pontecorvi@uniroma1.it

**Keywords:** retinopathy of prematurity, miRNAs, angiogenesis, retinal vascularization, VEGF signaling, oxygen-induced retinopathy, inflammation, retinal neovascularization, non-invasive biomarkers, anti-VEGF therapy

## Abstract

Retinopathy of Prematurity (ROP), a leading cause of blindness in preterm infants, arises from dysregulated angiogenesis and inflammation. Without timely intervention, ROP can progress to severe outcomes, including dense fibrovascular plaques and retinal detachment. MicroRNAs (miRNAs) regulate key pathways such as hypoxia response, VEGF signaling, and vascular remodeling. Studies have identified miRNAs (e.g., miR-210, miR-146a, and miR-21) as potential biomarkers and therapeutic targets. Preclinical evidence supports miRNA-based therapies (e.g., miR-18a-5p and miR-181a), targeting HIF-1α and VEGFA to mitigate neovascularization, with nanoparticle delivery systems enhancing stability and specificity. These strategies, combined with anti-VEGF agents, show significant potential for improving ROP management. While promising, miRNA therapies require validation in clinical trials to ensure safety and efficacy. This review discusses the role of miRNAs in ROP, highlighting their relevance as diagnostic and therapeutic tools.

## 1. Introduction

Retinopathy of Prematurity (ROP) is a proliferative retinal disorder affecting preterm infants, characterized by retinal ischemia, pathological neovascularization, and, in severe cases, retinal detachment and blindness [1]. ROP remains a leading cause of preventable vision loss in preterm infants, particularly those born at lower gestational ages or with reduced birth weights [2,3]. The underlying pathophysiology involves dysregulated angiogenesis driven by hypoxia-induced factors such as Vascular Endothelial Growth Factor (VEGF). Other angiogenic mediators, including insulin-like growth factor (IGF-I), angiotensin-converting enzyme (ACE), fibroblast growth factor (bFGF), tumor necrosis factor-alpha (TNF-α), and nitric oxide (NO), also play crucial roles in disease progression [4].

Emerging evidence suggests that microRNAs (miRNAs) are pivotal in retinal vascular development and disease mechanisms [5,6]. MiRNAs are small, non-coding RNAs that regulate gene expression post-transcriptionally by targeting mRNAs, influencing cellular processes such as proliferation, differentiation, and apoptosis [7]. Highly expressed in endothelial cells, miRNAs help to balance inhibitory and activating angiogenic factors, thereby contributing to vascular integrity and the regulation of angiogenesis. Studies in animal models and limited human investigations have identified specific miRNAs as potential markers of ROP progression and severity [8,9].

This review examines the current evidence on miRNA expression in ROP, focusing on their role as biomarkers and potential therapeutic targets.

## 2. Pathogenesis Classification and Treatment of ROP

Retinopathy of Prematurity (ROP) is a multifactorial disease affecting premature infants with incomplete or abnormal retinal vascularization [10]. It arises when normal retinal blood vessel growth (vasculogenesis) is disrupted, leading to pathological new vessel proliferation (angiogenesis). This abnormal proliferation occurs at the junction between vascularized and avascular retina, with vessels extending into the vitreous cavity. Without timely intervention, ROP can progress to severe complications, such as dense fibrovascular plaques behind the lens and complete tractional retinal detachment [11]. Historically referred to as retrolental fibroplasia, ROP remains one of the leading causes of childhood blindness worldwide [12].

Prematurity and low birth weight are the primary risk factors for ROP. Premature birth, defined as delivery before 37 weeks of gestation, is particularly associated with retinal disease in infants born before 32 weeks [2]. Advances in neonatal care have increased the survival rates of preterm infants, particularly in resource-limited settings, but have also led to a corresponding rise in ROP incidence. In the United States, the National Institutes of Health estimates that 1100 to 1500 infants require treatment for ROP annually, with 400 to 600 at a risk of severe visual impairment [3].

### 2.1. Pathogenesis

ROP develops in two distinct phases, primarily driven by fluctuations in oxygen levels in the developing retina [11] (Figure 1).

Phase 1 (hyperoxic phase): From birth to approximately 31 weeks’ gestational age, the retina is exposed to relative hyperoxia compared to the in-utero environment. This suppresses the production of VEGF and IGF-1, two essential molecules for normal retinal vascular development, resulting in halted blood vessel growth and avascular retinal regions [13].Phase 2 (hypoxic phase): Between 31- and 34-weeks’ gestational age, the metabolic demands of the retina increase, causing relative hypoxia in the avascular areas. This triggers excessive VEGF production, leading to disorganized and abnormal vascular proliferation. These fragile new vessels are prone to hemorrhage and contribute to tractional forces on the retina, potentially resulting in retinal detachment [14].

The pathophysiology of ROP is closely linked to oxygen management in neonatal intensive care. Historical evidence from the 1950s established a connection between excessive oxygen supplementation and severe ROP, prompting significant changes in neonatal care practices. Despite advancements, precise oxygen regulation remains essential to reducing the risk of severe ROP [15].

**Figure 1 biomedicines-13-00400-f001:**
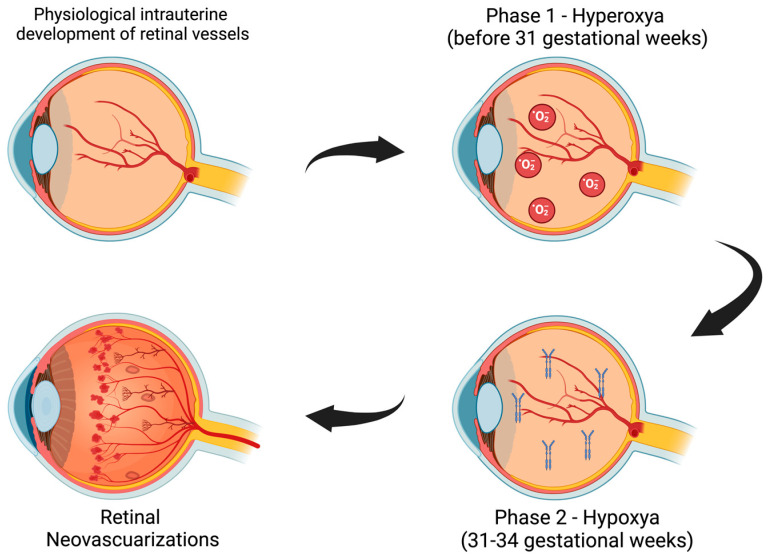
Schematic representation of the development of retinal neovessels in Retinopathy of Prematurity. During the intrauterine stage, vessels develop normally, but in the case of preterm birth, vascularization remains incomplete. Compared to the intrauterine stage, from birth to the 31st gestational week, the retina is exposed to relative hyperoxia, leading to the inhibition of VEGF and IGF-1. However, between 31 and 34 weeks, as the metabolic demand of the retina increases, it becomes exposed to relative hypoxia, triggering excessive VEGF production and the formation of neovessels [16].

### 2.2. Classification

ROP is classified according to the International Classification of Retinopathy of Prematurity (ICROP), first introduced in 1984 and updated in 2005 and 2021 [17]. This system categorizes ROP based on four key features:Location (Zones) as shown in Figure 2:
▪Zone I: The posterior retina within a circle centered on the optic nerve.▪Zone II: Extends from the edge of Zone I to the nasal ora serrata.▪Zone III: The outermost crescent of the retina.
Figure 2Schematic representation of the retinal zones used to classify the location of Retinopathy of Prematurity (ROP) according to the International Classification of Retinopathy of Prematurity (ICROP). The zones are defined as follows: Zone I represents the posterior retina within a circle centered on the optic nerve; Zone II extends from the edge of Zone I to the nasal ora serrata; and Zone III forms the outermost crescent of the retina.
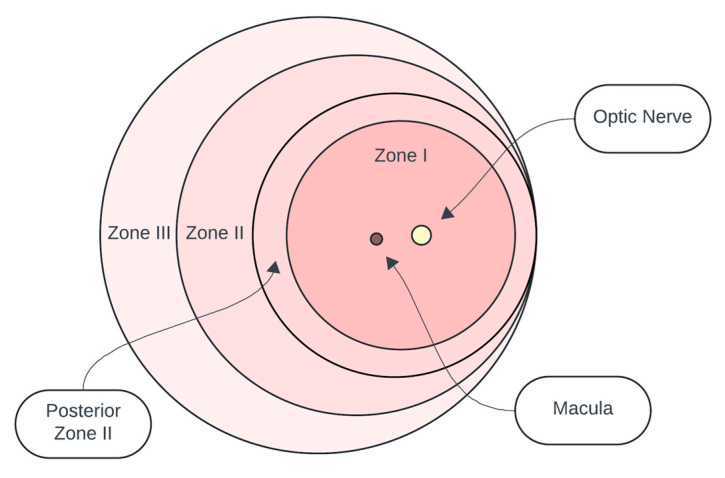

2.Extent of disease: Quantified in clock hours (1 to 12), indicating the proportion of the retina affected.3.Severity (Stages):
▪Stage 0: Immature retinal vasculature without pathological changes.▪Stage 1: A thin white demarcation line separating vascularized from avascular retina.▪Stage 2: The demarcation line develops height, width, and volume (the “ridge”), possibly with popcorn-like tufts of neovascular tissue.▪Stage 3: Fibrovascular proliferation extends from the ridge into the vitreous.▪Stage 4: Partial retinal detachment.▪Stage 5: Complete retinal detachment.
4.Plus disease: Characterized by dilated and tortuous retinal vessels in Zone I, often accompanied by iris vascularization and vitreous haze, indicating more severe disease (Figure 3).5.Additional terminology is used to describe disease severity [18]:
▪Threshold ROP: Defined as at least five contiguous or eight cumulative cloc hours of Stage 3 disease with plus disease, associated with a 50% risk of unfavorable outcomes if untreated.▪Pre-threshold ROP: Subdivided into the following:
○Type 1: Requiring treatment.○Type 2: Requiring close monitoring.


## 3. Development of Retinal Vascularization

Retinal vascularization is a highly regulated and complex process essential for delivering oxygen and nutrients to the inner retina, supporting the development and function of retinal neurons [1,19]. As one of the most metabolically active tissues in the body, the retina relies on precise vascular development to maintain normal vision.

The formation of retinal vasculature has been extensively studied using animal models, particularly mouse models, which replicate many aspects of human retinal vascular development [20]. These studies have significantly advanced our understanding of the signaling pathways and cellular interactions involved in this process under both physiological conditions and pathological scenarios, such as ROP and oxygen-induced retinopathy (OIR).

### 3.1. Anatomy of Retinal Vascularization

The retina comprises two distinct vascular systems: the retinal vasculature, which supplies the inner retina, and the choroidal vasculature, which nourishes the outer retina, including the photoreceptors. During retinal vascularization, capillaries penetrate the inner retina, forming a laminar meshwork, while the outer retina remains avascular [21]. The development of the retinal vasculature follows a highly organized pattern, originating from the optic nerve head and progressing outward to establish distinct vascular plexuses: the superficial, intermediate, and deep plexuses [22,23,24].

Superficial plexus: The first plexus forms at the inner surface of the retina. Vessels emerge from the optic nerve head and spread radially across the nerve fiber layer.Intermediate and deep plexuses: Angiogenic sprouts from the superficial plexus penetrate deeper retinal layers, forming secondary networks within the inner retina.

In humans, retinal vascular development begins in utero, whereas in mice, it occurs postnatally, making the mouse model a valuable tool for studying the intricate mechanisms of retinal vascularization.

### 3.2. Stages of Retinal Vascular Development

Hyaloid Vasculature Formation and Regression

Retinal vascular development begins with the formation of the hyaloid vasculature, a temporary vascular system that nourishes the developing eye during the embryonic stage. Originating from the hyaloid artery, this network supplies the retina and the posterior lens. As the retina matures, the hyaloid vessels regress, allowing retinal vessels to grow from the optic nerve head into the retina.

The regression of the hyaloid system, a process critical for normal eye development, starts around 13 weeks of gestation in humans and is typically complete by birth. The dysregulation of this regression can result in persistent fetal vasculature, a condition associated with visual impairment [25].

II.Formation of the Retinal Vascular Plexuses

The primary retinal vascular plexus develops through angiogenesis, the sprouting of new vessels from pre-existing ones. This process begins at the optic nerve head and extends centrifugally toward the retinal periphery. By approximately 36 weeks of gestation, the primary vascular network reaches the nasal periphery, and, by 40 weeks, the temporal periphery is fully vascularized [26].

Following the formation of the superficial plexus, angiogenic sprouts penetrate deeper retinal layers to establish two additional capillary layers: the intermediate and deep plexuses. These networks grow in a similar outward pattern as the superficial plexus but extend deeper into the retina. In full-term infants, the retinal vasculature is typically complete by birth.

## 4. Molecular Mechanisms in Retinal Angiogenesis

### 4.1. VEGF Signaling

Vascular Endothelial Growth Factor (VEGF) plays a critical role in retinal vascular development. Retinal astrocytes, which precede the formation of blood vessels, secrete VEGF in response to physiological hypoxia. VEGF promotes endothelial cell proliferation, migration, and survival, guiding vessel growth toward the hypoxic, avascular retina [27,28].

VEGF signaling is tightly regulated to ensure proper retinal vascularization. Disruptions in VEGF pathways, as demonstrated in oxygen-induced retinopathy (OIR) models, can lead to either excessive vessel growth (neovascularization) or vessel regression, contributing to conditions such as ROP [29].

### 4.2. PDGF and Retinal Astrocytes

Platelet-Derived Growth Factor (PDGF) is another essential signaling molecule in retinal vascular development. Retinal astrocytes, which express PDGF receptor alpha (PDGFRA), migrate across the inner retinal surface in response to PDGF produced by retinal ganglion cells (RGCs) [30,31,32]. This migration establishes a scaffold that guides developing blood vessels.

Astrocytes have a dual role in retinal vascularization: they secrete VEGF to stimulate vessel growth and provide a physical substrate for endothelial cell migration [33,34]. The loss of retinal astrocytes significantly impairs vascular development, resulting in abnormal or incomplete vascularization.

### 4.3. Neural-Vascular Interactions in Retinal Development

The retinal vasculature is closely regulated by neural signals, particularly those from RGCs. RGCs detect hypoxia and respond by producing VEGF, aligning the metabolic demands of the developing retina with oxygen and nutrient supply from the growing vascular network [35].

RGCs also produce anti-angiogenic factors, such as semaphorin 3E (SEMA3E), which help to restrict blood vessel growth to appropriate retinal layers, ensuring proper vascular lamination [36]. The dysregulation of this neural-vascular communication can result in misdirected vessel growth, contributing to retinal vascular pathologies.

## 5. Current Treatment for ROP

The management of ROP has undergone significant advancements since the late 20th century. The landmark Cryotherapy for Retinopathy of Prematurity (CRYO-ROP) study in the 1980s demonstrated that ablating the avascular peripheral retina could reduce the risk of severe visual impairment in eyes with threshold ROP [37]. However, nearly 25% of treated eyes still experienced unfavorable outcomes, particularly in cases involving posterior disease.

In the 1990s, laser photocoagulation replaced cryotherapy as the preferred treatment for ROP. Laser therapy, delivered via an indirect ophthalmoscope, enables the precise ablation of the peripheral retina and remains the standard care for severe ROP. The Early Treatment for Retinopathy of Prematurity (ETROP) study refined treatment protocols, recommending earlier intervention for high-risk pre-threshold ROP (type 1 ROP) [38]. Type 1 ROP, characterized by Zone I disease with plus disease or Stage 3 ROP, requires immediate treatment, while type 2 ROP necessitates close monitoring.

Over the past decade, anti-VEGF agents have emerged as a promising alternative or adjunct to laser therapy for ROP. These agents target VEGF, a critical mediator of pathological neovascularization in ROP, and have been widely used in adult ophthalmic conditions. Anti-VEGF therapy offers a more targeted approach to reducing abnormal vessel proliferation without the extensive retinal damage associated with laser treatment.

The landmark BEAT-ROP trial was the first large-scale study to evaluate anti-VEGF therapy in ROP, specifically investigating intravitreal bevacizumab [39]. The study demonstrated a significant reduction in ROP recurrence in Zone I disease compared to laser therapy. However, concerns regarding bevacizumab’s systemic effects, including potential impacts on neurodevelopment due to prolonged VEGF suppression, remain. VEGF is essential for normal infant neurodevelopment, and systemic suppression following bevacizumab treatment has raised safety concerns.

The RAINBOW trial compared intravitreal ranibizumab (0.1 mg and 0.2 mg doses) to laser therapy in a randomized, multicenter study across 26 countries [40]. Ranibizumab, a smaller anti-VEGF molecule with a shorter systemic half-life than bevacizumab, potentially minimizes systemic VEGF suppression. The trial found that ranibizumab 0.2 mg achieved higher treatment success rates than laser therapy, with fewer structural complications and a reduced incidence of high myopia. Additionally, systemic VEGF levels were not suppressed, addressing key safety concerns. Despite these short-term benefits, questions remain about the long-term safety and efficacy of anti-VEGF therapy [41]. The RAINBOW Extension Study, which followed infants for up to five years, confirmed that ranibizumab 0.2 mg maintained superior ocular outcomes compared to laser therapy, including fewer cases of high myopia and no new ocular complications. Visual acuity outcomes were comparable between the groups, with slightly better vision-related quality of life in the ranibizumab group. Importantly, the study found no significant differences in neurodevelopmental outcomes or systemic health between the two treatment groups [42].

Anti-VEGF agents present several advantages over laser therapy, including shorter procedural times, reduced anesthesia requirements, and fewer visual field defects. However, these benefits come with the trade-off of potential disease recurrence, which can occur later than with laser therapy. Consequently, infants treated with anti-VEGF agents require prolonged and more frequent follow-up to monitor for late recurrences and ensure complete retinal vascularization. Intravitreal ranibizumab, now licensed in Europe for ROP treatment, offers a viable alternative to laser therapy, particularly in posterior disease or cases at high risk for poor outcomes. Clinicians must balance the improved ocular outcomes and reduced risk of myopia against the need for extended follow-up and the uncertainty surrounding potential systemic risks [43].

Further research is necessary to optimize dosing, evaluate long-term systemic safety, and clarify the role of anti-VEGF agents in combination with laser therapy. Ongoing studies, such as the RAINBOW Extension Study, will continue to provide critical insights into the long-term outcomes of anti-VEGF therapy in ROP.

## 6. ROP and miRNA

MicroRNAs (miRNAs) are small, single-stranded, non-coding RNA molecules typically 18–26 nucleotides in length [44]. They are transcribed by RNA polymerase II as primary transcripts (pri-miRNAs) containing hairpin structures. These are processed in the nucleus by the Microprocessor complex (Drosha–DGCR8), generating precursor miRNAs (pre-miRNAs). The pre-miRNAs are then exported to the cytoplasm by exportin-5 and cleaved by Dicer into ~22-nucleotide duplexes. The guide strand is loaded into an Argonaute-containing RISC (RNA-induced silencing complex) to regulate gene expression, either through mRNA degradation or translational repression. Each miRNA can target multiple transcripts, thereby influencing extensive gene networks and participating in feedback loops with epigenetic mechanisms, such as DNA methylation and histone modifications [45]. These molecules play an important role in the post-transcriptional regulation of gene expression by binding to complementary sequences in the 3′ untranslated regions (UTRs) of target messenger RNAs (mRNAs). This binding leads either to the degradation of the target mRNA or the inhibition of its translation, thus controlling the expression of various proteins involved in cellular processes like apoptosis, differentiation, immune regulation, and angiogenesis [46].

In the context of Retinopathy of Prematurity (ROP), the genetic and environmental complexity underscores the role of miRNAs as central regulators, particularly since no specific mutations in relevant genes have been identified [47]. While large-scale whole-exome sequencing (WES) analyses have revealed potentially relevant pathways, they have not identified any single gene with genome-wide significance [48]. Similarly, proteomic studies—such as those analyzing tear fluid exosomes—have identified key proteins associated with ROP pathology such as Cu/Zn-superoxide dismutase [49]. However, these findings primarily reflect downstream pathway activation rather than the upstream regulatory mechanisms driving disease progression. Given their relative stability and role as “master regulators,” miRNAs offer a distinct advantage by modulating multiple genes and pathways simultaneously at both transcriptional and post-transcriptional levels. Consequently, miRNAs hold strong potential to address the multifactorial nature of ROP by bridging genetic and environmental factors, making them promising candidates for both biomarker development and targeted therapies.

### 6.1. Role of miRNA in Angiogenesis and Retinopathy of Prematurity

Angiogenesis, the process through which new blood vessels form from pre-existing vasculature, is essential for normal retinal development [50,51]. However, in pathological conditions such as ROP, dysregulated angiogenesis leads to abnormal blood vessel proliferation and subsequent retinal damage.

MicroRNAs (miRNAs) are critical regulators of angiogenesis, influencing the balance between pro- and anti-angiogenic signals. Studies on animal models of ROP have identified several miRNAs involved in the pathogenesis of the disease (Figure 4):▪miR-210: Hypoxia-induced miR-210 plays a key role in cellular responses to low oxygen levels. Its downregulation in ROP is associated with disease progression as it counteracts hypoxic stress and reduces vascular proliferation [52,53,54].▪miR-21: Known for promoting cell survival and vascular proliferation, miR-21 contributes to the fibrovascularization process in advanced ROP stages. Its overexpression is linked to abnormal vessel growth [54,55,56].▪miR-146a: This miRNA modulates endothelial cell activity and inhibits angiogenesis through interaction with nuclear factor kappa B (NF-kB). Its downregulation in ROP leads to unchecked vascular remodeling [54,57,58,59].▪miR-221 and miR-222: These miRNAs inhibit endothelial cell proliferation and migration by targeting angiogenesis-related pathways. Their dysregulation exacerbates abnormal neovascularization in ROP [60].▪miR-23a and miR-200b-3p: Both are upregulated in hyperoxia-induced rat models of ROP. miR-23a may protect retinal cells under oxidative stress, while miR-200b-3p regulates VEGF-A expression, a key angiogenic factor [51,61,62].▪miR-27b-3p and miR-214-3p: These miRNAs are downregulated in ROP, leading to increased expression of pro-angiogenic factors such as VEGF-B and VEGF-C, contributing to pathological neovascularization [60,63].▪miR-143 and miR-126: Involved in retinal neovascularization, these miRNAs exhibit roles in promoting or inhibiting angiogenesis depending on the specific conditions [6].

The balance of these miRNAs is crucial in determining whether angiogenesis is appropriately regulated or leads to pathological outcomes, as seen in ROP.

Recent bioinformatics analyses have also highlighted the roles of additional miRNAs in ROP, including the following:▪miR-128-3p: Upregulated in ROP, it is associated with angiogenesis and cell migration, regulating pathways such as PI3K-Akt and TGF-β signaling, both critical for vascular development [64].▪miR-9-5p: Downregulated in ROP, this miRNA is primarily involved in neural development but also influences retinal vascular disorders, suggesting a neurovascular regulatory role in abnormal angiogenesis [65].

Investigating miRNA expression profiles and their associated molecular targets has provided valuable insights into the complex mechanisms underlying ROP pathogenesis. These findings may guide future therapeutic strategies targeting specific miRNAs to restore angiogenic balance and prevent disease progression.

**Figure 4 biomedicines-13-00400-f004:**
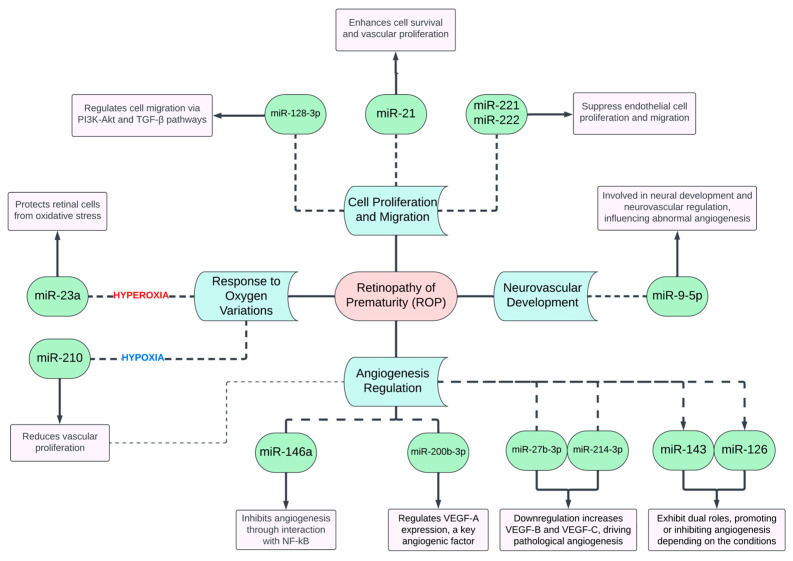
Schematic representation of the roles of key miRNAs in Retinopathy of Prematurity (ROP).

### 6.2. Plasma miRNA Profiles in ROP

The plasma profiling of infants with ROP has identified dysregulated miRNA expression patterns associated with disease progression. Elevated levels of miR-23a and miR-200b-3p have been linked to pro-angiogenic processes, whereas miR-27b-3p and miR-214-3p exhibit anti-angiogenic activity [61].

Ovali et al. demonstrated significant alterations in the expression profiles of specific miRNAs, including miR-210, miR-146a, miR-21, miR-143, miR-221, and let-7, in preterm infants with ROP compared to those without the disease. Notably, miR-210 levels were reduced in the early stages of ROP but increased during later stages, highlighting a dynamic role in disease progression [54].

These plasma-derived miRNA signatures offer valuable insights into the molecular mechanisms of ROP and hold promise as non-invasive diagnostic and prognostic biomarkers for the condition (Table 1).

### 6.3. Experimental Models and miRNA Regulation

Animal models of ROP, particularly oxygen-induced retinopathy (OIR) in mice, have been essential for understanding the functional roles of miRNAs in retinal neovascularization. Hyperoxia-induced ROP models have shown differential expression of 22 upregulated and 44 downregulated miRNAs. For instance, miR-9a-5p was activated, while miR-223-3p was suppressed, influencing VEGF signaling and inflammatory responses [55,65]. Other miRNAs, such as miR-301a, miR-130a, and miR-128-3p, have also been identified in these models, providing insights into their roles in the pathogenesis of ROP [66]. The administration of miR-18a-5p in a mouse OIR model suppressed retinal neovascularization by targeting pro-angiogenic factors such as FGF1 and HIF1A [67]. Additionally, miR-301a and miR-130a were found to regulate peroxisome-proliferator-activated receptor gamma (PPARγ), a transcription factor involved in ocular angiogenesis [66].

These experimental models have contributed to the identification of key miRNAs and clarified their mechanistic roles, offering a basis for the development of miRNA-based therapeutic strategies.

## 7. Current Applications of miRNAs in Diagnosis and Therapy

MiRNAs are emerging as non-invasive biomarkers for various diseases due to their stability in biological fluids such as blood, urine, and saliva [68]. Their expression profiles can reflect specific disease states, making them invaluable for early diagnosis and monitoring. Several diagnostic tests leveraging miRNA expression are already available to clinicians. For example, miRview™ Mets is used to identify the tissue of origin in cancers of unknown primary, achieving 90% accuracy in determining tumor origin and enabling more precise, personalized treatment strategies [69]. Similarly, RosettaGX Reveal, a diagnostic tool for thyroid cancer, utilizes miRNA panels to differentiate between benign and malignant lesions with high sensitivity and specificity [70]. OsteomiR has been designed to assess fracture risk in postmenopausal women and patients with type 2 diabetes based on miRNA profiles [71]. Additionally, circulating miRNA panels are being investigated for the early detection of neurodegenerative diseases such as Alzheimer’s, as well as for monitoring treatment responses in cancers and autoimmune diseases [72].

In recent years, miRNA-based therapies have made significant progress, with several approaches currently undergoing clinical trials. These advancements highlight the potential of miRNA molecules as therapeutic tools for treating a wide range of diseases, including fibrotic disorders, cancer, viral infections, and inflammatory conditions. One of the most promising strategies involves miRNA mimics, which are synthetic molecules designed to restore the function of downregulated miRNAs [73]. For instance, miR-29 mimics are being developed as potential treatments for fibrotic diseases, while miR-34a mimics, formulated within nanoparticles, are being evaluated for their efficacy against various types of cancer [74,75]. Another key approach is the use of AntagomiRs, chemically modified antisense oligonucleotides that inhibit overexpressed miRNAs [76]. Notable examples include Miravirsen, an anti-miR-122 molecule that has completed phase II clinical trials for hepatitis C treatment [77]. Additionally, RG-012 (anti-miR-21) is being investigated as a potential therapy for fibrotic diseases such as Alport syndrome, while anti-miR-155 is being widely investigated in oncology [78,79]. Beyond these direct miRNA-targeting approaches, researchers are also exploring small molecule modulators that regulate miRNA expression. One such example is ABX464, a drug that induces the expression of miR-124 and is in phase II trials for the treatment of Crohn’s disease and ulcerative colitis [80]. As these therapeutic strategies continue to advance through clinical development, they hold great promise for expanding the treatment landscape of numerous diseases, potentially offering novel and more targeted therapeutic options in the near future [68].

### Therapeutic Potential of miRNAs in ROP

The ability of miRNAs to modulate critical pathways involved in retinal angiogenesis and inflammation makes them attractive therapeutic targets for ROP.

Recent studies have demonstrated the potential of miRNA-based therapies in animal models of ROP. For example, the delivery of synthetic miRNA mimics, such as miR-18a-5p, has been explored as a potential therapeutic strategy. The intravitreal injection of miR-18a-5p mimic in a mouse model of oxygen-induced retinopathy effectively suppressed retinal neovascularization by targeting key pro-angiogenic factors, FGF1 and HIF1A [81]. MiR-181a inhibits VEGFA expression directly, thereby impairing endothelial cell migration and tube formation in models of oxygen-induced retinopathy [82].

Nanoparticle-mediated delivery systems have further enhanced the stability, bioavailability, and targeted delivery of miRNA-based therapies [67]. These systems protect miRNAs from enzymatic degradation and improve their retention at target sites, such as the retina. Studies using animal models have shown that nanoparticles carrying miRNA mimics can penetrate retinal layers, enabling sustained release and prolonged therapeutic effects [83,84,85].

Combination therapies integrating miRNA-based treatments with anti-VEGF agents have demonstrated synergistic effects, effectively reducing vascular leakage and inflammatory responses beyond what either therapy alone [86]. The interplay between hypoxia-associated miRNAs (hypoxamiRs) and the VEGF pathway represents a promising therapeutic avenue for ROP as preclinical models suggest that miRNA-based strategies may enhance the efficacy of anti-VEGF treatments in mitigating pathological angiogenesis and vascular instability. These innovative approaches, which harness the regulatory functions of miRNAs, could potentially overcome the limitations of current monotherapies. However, to date, no in vivo or clinical studies have validated the therapeutic role of miRNAs in ROP, and no miRNA-based treatments are currently available for use in humans.

## 8. Conclusions

MicroRNAs seem to be central to the pathogenesis of ROP, influencing angiogenesis, inflammation, and hypoxia-driven pathways. Their dual role as biomarkers and therapeutic targets presents new opportunities for advancing diagnosis and treatment. Preclinical studies highlight the potential of miRNA-based therapies, particularly through nanoparticle-mediated delivery systems that enhance stability and targeting precision.

Despite their promise, the clinical application of miRNA therapies requires further validation in large-scale trials. Future research should prioritize optimizing delivery strategies, integrating miRNA therapies with current anti-VEGF and laser treatments, and ensuring long-term safety. By leveraging miRNAs’ regulatory potential, it may be possible to revolutionize ROP management, offering more tailored and effective interventions to mitigate vision loss in premature infants.

## Figures and Tables

**Figure 3 biomedicines-13-00400-f003:**
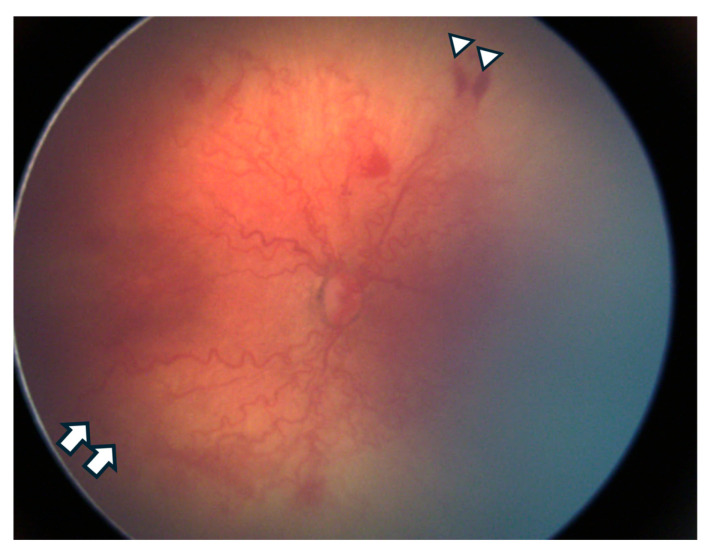
Retinography of a premature infant with aggressive Retinopathy of Prematurity (A-ROP), demonstrating marked vascular tortuosity characteristic of plus disease, vascular loops (white arrows), and flat extraretinal vascularization (white arrowheads). A-ROP, as defined in the third edition of the International Classification of ROP (ICROP3), includes rapidly progressing forms such as aggressive posterior ROP (AP-ROP), which can bypass the typical staged progression of the disease.

**Table 1 biomedicines-13-00400-t001:** Dysregulated plasma miRNA expression profiles and their functional effects in retinopathy of prematurity.

miRNA	Expression	Effect
miR-210	↓	Hypoxia response [52,53,54]
miR-146a	↓	Inflammation [54,57,58,59]
miR-21	↑	Angiogenesis [54,55,56]
miR-143	↓	Angiogenesis regulation [6]
miR-221	↑	Endothelial apoptosis [60]
miR-23a	↑	Oxidative stress protection [62]
miR-200b-3p	↑	VEGF modulation [51,61]
miR-27b-3p	↓	Anti-angiogenic [63]
miR214-3p	↓	Anti-angiogenic [60]

Upregulated miRNAs are marked as (↑), while downregulated miRNAs are marked as (↓).

## Data Availability

Not applicable.

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
