# Peer review of "Retinopathy of Prematurity and MicroRNAs"

_biomedicines, 2025, doi:10.3390/biomedicines13020400_

Round 1
Reviewer 1 Report
Comments and Suggestions for Authors
The manuscript entitled “Retinopathy of prematurity and microRNAs” shows interesting information for the treatment of retinopathy of prematurity (ROP) by using microRNAs (miRNAs) that target HIF-1α and VEGF in preterm infants. It includes pathogenesis classification and treatment of ROP, development of retinal vascularization, molecular mechanisms in retinal angiogenesis, ROP and miRNA, and therapeutic potential of miRNAs in ROP. Although this manuscript was prepared and written well with updated therapeutic methods of ROP, it still needs to be clarified for some points as follows:
1. Due to the title of the manuscript, the authors should inform more general details of miRNAs in this manuscript. For example, the biogenesis of miRNA in the cells, the roles of miRNAs in biological processes and cell functions, and the current applications of miRNA for treatment of some diseases apart from the treatment of ROP.
2. Could the authors compare the therapeutic effects of miRNAs to other treatment methods of ROP, such as anti-VEGF agents and lasers? The authors can show the current data that are reported in the publications from either in vivo experiments or clinical studies.
3. The sequence of contents in the manuscript should be rearranged. The section No. 3 (development of retinal vascularization) and the section No.4 (molecular mechanisms in retinal angiogenesis) should be presented before the section No. 2.3 (treatment).
4. The section No. 3 (development of retinal vascularization) should be described with a diagram or a picture to elucidate the anatomy of retinal vascularization and the stages of retinal vascular development.
5. The authors should add an arrow/arrows to point the lesions of ROP according to the figure caption of Figure 2.
Author Response
The manuscript entitled “Retinopathy of prematurity and microRNAs” shows interesting information for the treatment of retinopathy of prematurity (ROP) by using microRNAs (miRNAs) that target HIF-1α and VEGF in preterm infants. It includes pathogenesis classification and treatment of ROP, development of retinal vascularization, molecular mechanisms in retinal angiogenesis, ROP and miRNA, and therapeutic potential of miRNAs in ROP.
R: Dear reviewer, thank you for taking the time to read our manuscript and for your precious comments. Please find below our replies.
Although this manuscript was prepared and written well with updated therapeutic methods of ROP, it still needs to be clarified for some points as follows:
- Due to the title of the manuscript, the authors should inform more general details of miRNAs in this manuscript. For example, the biogenesis of miRNA in the cells, the roles of miRNAs in biological processes and cell functions, and the current applications of miRNA for treatment of some diseases apart from the treatment of ROP.
R: Thank you for your suggestion. We have added two paragraphs at the beginning of section 6 (ROP and mirRNA). And also a paragraph regarding the current applications of miRNA for different diseases. See lines: 274-303 and lines 407-427 in the tracked manuscript.
- Could the authors compare the therapeutic effects of miRNAs to other treatment methods of ROP, such as anti-VEGF agents and lasers? The authors can show the current data that are reported in the publications from either in vivo experiments or clinical studies.
R: Thank you for your comment. To date, there are no clinical applications of miRNA-based therapies for ROP, and direct comparisons remain premature. The available data are currently limited to preclinical studies, with only one in vivo study investigating the potential role of miRNA modulation in an ROP model. To address this, we have expanded the discussion at the beginning of section 7 (lines 394-406) and at the end of Section 7.1 (lines 445–457), incorporating the existing preclinical findings even for other diseases. While these data are still preliminary, they highlight the potential of miRNAs as future therapeutic targets. However, further research is needed before meaningful comparisons with established treatments can be made.
- The sequence of contents in the manuscript should be rearranged. The section No. 3 (development of retinal vascularization) and the section No.4 (molecular mechanisms in retinal angiogenesis) should be presented before the section No. 2.3 (treatment).
R: Thank you for your comment. We have rearranged the sections as you suggested and we agree that it results in a better structure.
- The section No. 3 (development of retinal vascularization) should be described with a diagram or a picture to elucidate the anatomy of retinal vascularization and the stages of retinal vascular development.
R: Thank you for your suggestion (interestingly, the other reviewer has made a similar suggestion). We have added a new figure showing the stage of development of retinal neovascularization showing the processes of hyperoxia and hypoxia according to the gestational age. See Figure 1.
- The authors should add an arrow/arrows to point the lesions of ROP according to the figure caption of Figure 2.
R: Thank you. As you suggested we added the arrows to show the lesions in the figure (now Figure 3).
Reviewer 2 Report
Comments and Suggestions for Authors
The review article entitled as “Retinopathy of Prematurity and MicroRNAs” explains the importance of miRNAs in disease pathogenesis and their relevance as diagnostic and therapeutic target. The authors described the molecular mechanism involved in retinal angiogenesis and its pathophysiology in association with ROP, and also highlighted the role of several miRNAs which regulate the key angiogenic pathways. The title of the study is conclusive and description in the review article is appropriate. A few minor suggestions:
· A visual information (figure) describing the stages of retinal vasculature development and different pathways which leads to pathophysiological angiogenesis would be a great addition to this review article.
· Table 1 should also include the fourth column with all the references for each miRNA expression profiles and their functional effects in ROP.
· A small paragraph describing about the genetics and proteomics study of ROP will aid why miRNAs should be the center of focus as a diagnostic and therapeutic target.
Author Response
The review article entitled as “Retinopathy of Prematurity and MicroRNAs” explains the importance of miRNAs in disease pathogenesis and their relevance as diagnostic and therapeutic target. The authors described the molecular mechanism involved in retinal angiogenesis and its pathophysiology in association with ROP, and also highlighted the role of several miRNAs which regulate the key angiogenic pathways. The title of the study is conclusive and description in the review article is appropriate.
R: Dear Reviewer, thank you very much for taking the time to read our manuscript and provide your valuable comments.
A few minor suggestions:
- A visual information (figure) describing the stages of retinal vasculature development and different pathways which leads to pathophysiological angiogenesis would be a great addition to this review article.
R: Thank you very much for your suggestion (interestingly, the other reviewer has made a similar suggestion). We have added a new figure showing the development of the retinal vasculature and the neovascularization (angiogenesis) according to the gestational age. See Figure 1.
- Table 1 should also include the fourth column with all the references for each miRNA expression profiles and their functional effects in ROP.
R: We have added the references as suggested.
- A small paragraph describing about the genetics and proteomics study of ROP will aid why miRNAs should be the center of focus as a diagnostic and therapeutic target.
R: Thanks to your suggestion we have expanded section 6 to include genetic and proteomic studies of ROP (see lines 290-303). We highlight that WES analyses have identified potential pathways but no single gene with genome-wide significance, while proteomic studies have identified key proteins reflecting downstream effects rather than upstream regulation. This reinforces the rationale for focusing on miRNAs, given their stability and role as master regulators integrating genetic and environmental factors.
Round 2
Reviewer 1 Report
Comments and Suggestions for Authors
The authors responded all comments and questions. They already revised the manuscript according to the suggestions. Therefore, this manuscript could be accepted for publishing in the journal in the present form.